# Beyond the WHO 2020 Classification of Female Genital Tumors: Types of Endometrial Cancer: A Pathological and Molecular Focus on Challenging Rare Variants

**DOI:** 10.3390/ijms251910320

**Published:** 2024-09-25

**Authors:** Angela Santoro, Giuseppe Angelico, Antonio Travaglino, Frediano Inzani, Damiano Arciuolo, Antonio d’Amati, Nicoletta D’Alessandris, Giulia Scaglione, Michele Valente, Belen Padial Urtueta, Francesca Addante, Nadine Narducci, Giuseppe Pannone, Emma Bragantini, Antonio Raffone, Antonino Mulè, Gian Franco Zannoni

**Affiliations:** 1Unità Operativa Complessa Anatomia Patologica Generale, Dipartimento di Scienze Della Salute Della Donna, del Bambino e di Sanità Pubblica, Fondazione Policlinico Universitario A. Gemelli IRCCS, Largo A. Gemelli 8, 00168 Roma, Italy; angela.santoro@policlinicogemelli.it (A.S.); damiano.arciuolo@policlinicogemelli.it (D.A.); antonio.damati@uniba.it (A.d.); nicoletta.dalessandris@policlinicogemelli.it (N.D.); giulia.scaglione@policlinicogemelli.it (G.S.); dr.valente.m@gmail.com (M.V.); belen.padialurtueta@policlinicogemelli.it (B.P.U.); francesca.addante@policlinicogemelli.it (F.A.); nadine.narducci@policlinicogemelli.it (N.N.); antonino.mule@policlinicogemelli.it (A.M.); 2Istituto di Anatomia Patologica, Università Cattolica del Sacro Cuore, Largo A. Gemelli 8, 00168 Roma, Italy; 3Department of Medical and Surgical Sciences and Advanced Technologies “G.F. Ingrassia”, Anatomic Pathology, University of Catania, 95100 Catania, Italy; giuangel86@hotmail.it; 4Pathology Unit, Department of Medicine and Technological Innovation, University of Insubria, 21100 Varese, Italy; antonio.travaglino@uninsubria.it; 5Anatomic Pathology Unit, Department of Molecular Medicine, University of Pavia and Fondazione IRCCS San Matteo Hospital, 27100 Pavia, Italy; frediano.inzani@unipv.it; 6Anatomic Pathology Unit, Department of Clinic and Experimental Medicine, University of Foggia, 71122 Foggia, Italy; giuseppe.pannone@unifg.it; 7Department of Surgical Pathology, Ospedale S. Chiara 9, 38122 Trento, Italy; emma.bragantini@apss.tn.it; 8Department of Medical and Surgical Sciences (DIMEC), University of Bologna, 40138 Bologna, Italy; anton.raffone@gmail.com; 9Gynecology and Obstetrics Unit, Department of Neuroscience, Reproductive Sciences and Dentistry, School of Medicine, University of Naples Federico II, 80131 Naples, Italy

**Keywords:** endometrial cancer, rare variants, histology, immunohistochemistry, molecular pathology, personalized medicine

## Abstract

Endometrial carcinoma is a heterogeneous group of malignancies characterized by distinct histopathological features and genetic underpinnings. The 2020 WHO classification has provided a comprehensive framework for the categorization of endometrial carcinoma. However, it has not fully addressed the spectrum of uncommon entities that are currently not recognized by the 2020 WHO and have only been described in the form of small case series and case reports. These neoplasms represent a real diagnostic challenge for pathologists; furthermore, their therapeutic management still remains controversial and information regarding tumor prognosis is very limited. This review aims to elucidate these lesser-known variants of endometrial carcinoma. We discuss the challenges of identifying these rare subtypes and the molecular alterations associated with them. Furthermore, we propose the need for expanded classification systems that include these variants to enhance clinical outcomes and research efforts. We believe that a better histological typing characterization of these entities may lead to more reproducible and accurate diagnoses and more personalized treatments. By raising awareness of these rare entities, we also hope to encourage further investigation and integration into clinical practice to improve patient care in endometrial carcinoma.

## 1. Introduction

Endometrial carcinoma is the sixth most frequently diagnosed cancer in women and the second most common carcinoma of the female genital tract [1]. The current World Health Organization (WHO) classification of tumors of the uterine corpus includes the common types of endometrial carcinoma, i.e., endometrioid carcinoma, serous carcinoma, clear cell carcinoma, carcinosarcoma, and undifferentiated/dedifferentiated carcinoma (UDC/DDC), as well as rare histotypes, i.e., mesonephric-like, gastrointestinal-type, and squamous cell carcinoma [2].

However, several other uncommon entities that are currently not recognized by the WHO have been described in the form of small case series and multiple case reports. Assessing these cases appears clinically relevant as they do not fit any of the current types of endometrial carcinoma and represent a challenge for pathologists. The differences between these histotypes and the other more common types of endometrial carcinoma, as well as their appropriate therapeutic approach, are still controversial. Moreover, information regarding tumor behavior and long-term prognosis is limited.

The purpose of this review is to present the clinicopathological and molecular features of these rare variants, discussing their differential diagnosis and management course. We believe that a better characterization of these entities may lead to more reproducible diagnoses and more accurate treatments.

## 2. Endometrial Giant Cell Carcinoma (EGCC)

Endometrial giant cell carcinoma (EGCC) is an uncommon endometrial carcinoma variant composed of a proliferation of extremely atypical, mono- or multinucleated giant cells [3,4,5].

These cells are arranged in poorly cohesive sheets and exhibit bizarre nuclear features, including multilobate nuclei, clear chromatin with eosinophilic nucleoli or hyperchromasia (often observed in different nuclei within the same cell), cytoplasmic intranuclear pseudoinclusions, and atypical mitotic figures. The giant cells are typically accompanied by a conventional carcinoma component, which is of endometrioid type in the vast majority of cases [5].

Immunohistochemically, the giant cells show partial loss of epithelial differentiation, with only focal cytokeratins and E-cadherin expression. Moreover, any marker of specific differentiation lineage is negative, including histiocytic markers, smooth muscle markers, striated muscle markers, neurogenic markers, and melanocytic markers. Regarding The Cancer Genome Atlas (TCGA) molecular classification, most cases fall into the “no specific molecular profile” (NSMP) group as they show wild-type p53 expression pattern, retained mismatch repair (MMR) proteins expression, and no polymerase epsilon (*POLE)* mutations [3,4,5,6,7,8,9,10].

The classification of EGCC is subject of debate. In fact, giant tumor cells can be observed in other more common histotypes of endometrial carcinoma, such as serous carcinoma and high-grade endometrioid carcinoma. However, the giant cells of EGCC appear more prominent in terms of both quantity (they are extensively present) and quality (they are extremely bizarre). Moreover, the giant cells of EGCC show loss of epithelial differentiation, as discussed above [5].

Abbas Agaimy suggested that giant cell carcinoma (not only endometrial but of any district) is not a distinct entity but a morphologic pattern in the spectrum of undifferentiated and sarcomatoid carcinoma [11]. Although we agree with such a view, we would remark that EGCC differs from the typical endometrial UDC/DDC and carcinosarcoma.

In fact, EGCC appears unrelated to classical UDC/DDC, which is typically composed of monomorphic undifferentiated cells and often shows mismatch repair (MMR)-deficiency and SWI/SNF proteins loss of expression [2,12,13]. The difference between EGCC and UDC/DDC is also acknowledged by the international society of gynecological pathologists [14].

On the other hand, carcinosarcoma most often shows a serous carcinoma component and is of the p53-abnormal group in the vast majority of cases [15]. A recent study suggested that carcinosarcoma is almost exclusively p53-abnormal [16]. Moreover, markers of specific mesenchymal differentiation are negative in EGCC [3,4,5].

Based on the available literature data, EGCC exhibits an aggressive behavior and is associated with advanced stage, lymph-node involvement, recurrences, and distant metastases [6,9]. It appears, therefore, reasonable to consider EGCC analogous to UDC/DDC and carcinosarcoma in terms of treatment [3,4,5].

### Summary Tips for Histological Diagnosis

EGCC is characterized by sheets of giant anaplastic, mono- and multinucleated cells with extreme pleomorphism.

The immunohistochemical profile is characterized by focal/multifocal CKAE1/AE3 and EMA expression, with loss of E-cadherin and absence of specific differentiation markers (such as β-HCG and myogenic markers).

In Table 1, histological, immunohistochemical, and molecular features are summarized, along with the relative references.

## 3. Pilomatrix-like High-Grade Endometrioid Carcinoma (PiMHEC)

Pilomatrix-like high-grade endometrioid carcinoma (PiMHEC) has been described by Weisman et al. as a variant of FIGO G3 endometrioid carcinoma morphologically resembling pilomatrix carcinoma, a malignant tumor of hair matrix cell origin [29]. Arciuolo et al. and Santoro et al. contributed to defining the distinctive features of PiMHEC [30,31].

Histologically, PiMHEC is characterized by solid nests of high-grade basaloid cells exhibiting ghost cell keratinization, similar to that observed in hair matrix tumor. Geographic necrosis is commonly found. The solid component of PiMHEC often coexists with a separate endometrioid component, which supports the endometrioid lineage of the tumor. Immunohistochemically, the basaloid cells show diffuse nuclear accumulation of β-catenin) which reflects the presence of an underlying *CTNNB1* mutation and is typically accompanied by CDX2 (caudal-type homeobox transcription factor 2) positivity. Interestingly, the tumor is negative for PAX8 (paired box gene 8) and estrogen and progesterone receptors, suggesting a loss of Müllerian differentiation; these markers are positive in the endometrioid component, if present. The expression of CK7 (cytokeratin 7), as well as of basal/squamous cell markers p63 and p40, is variable. A multifocal expression of neuroendocrine markers chromogranin and synaptophysin is frequently observed. PiMHEC is typically p53-wild-type; MMR-deficiency has been described in a subset of cases [29,30,31].

The question about whether PiMHEC should be considered as a distinct entity or as a morphological variant of FIGO G3 endometrioid carcinoma appears clinically relevant. In our opinion, there are several aspects that support the idea that PiMHEC is in fact a distinct entity. First of all, PiMHEC shows close resemblance to a neoplasm of another site, i.e., pilomatrix carcinoma. Immunohistochemistry highlights the loss of Müllerian differentiation, as discussed above. Moreover, data from the published studies suggest that PiMHEC may be significantly more aggressive than FIGO G3 endometrioid carcinoma. In the series by Weisman et al., all patients died of disease within 2 years from diagnosis [29]. This might indicate the need for a more aggressive treatment compared to endometrioid carcinoma. Finally, PiMHEC shows some distinctive and consistent features that may allow a reliable differential diagnosis with its mimickers. In our previous study, we tried to elaborate diagnostic criteria for PiMHEC; these include: solid nests of high-grade basaloid cells with ghost cell keratinization; the solid nests should not be admixed with a glandular component (a glandular component can only be present as a separate component); diffuse nuclear β-catenin accumulation in the solid nests (no areas of membranous expression); loss of PAX8 and hormone receptors expression in the solid nests [31].

### 3.1. Mimickers of PiMHEC Include

High-grade endometrioid carcinoma with a solid pattern and squamous features; these tumors usually lack ghost cell keratinization, do not show diffuse nuclear β-catenin accumulation, and show at least partially retained expression of Müllerian markers [31];

Low-grade endometrioid carcinoma with prominent ghost cell keratinization; keratinization in these cases is typically associated with morular metaplasia; there is an admixture of squamous and glandular elements, which is also highlighted immunohistochemically (mixed membranous and nuclear β-catenin expression; patchy positivity for Müllerian markers) [31];

Serous carcinoma with a solid growth pattern; this tumor is p53-mutant and does not show keratinization [2,14,42];

UDC/DDC with diffuse nuclear β-catenin accumulation and areas of “abrupt keratinization”; in these cases, the undifferentiated cells are dyscohesive and organized in “patternless” sheets; no prominent ghost cell keratinization is usually observed [2,14,31,42];

Neuroendocrine carcinoma; this tumor is diffusely positive for neuroendocrine markers (which can be multifocally positive in PiMHEC) and does not show ghost cell keratinization [31,42];

Squamous cell carcinoma; this is rare in the endometrium and not associated with nuclear β-catenin accumulation or ghost cell keratinization [2,43].

Further studies are warranted in this field. In Table 2, mimickers of PiMHEC and their features are summarized.

### 3.2. Summary Tips for Histological Diagnosis

PiMHEC is characterized by a solid proliferation of high-grade basaloid cells, associated with diffuse ghost cell keratinization and extensive geographic necrosis. In most cases, PiMHEC exhibits a low-grade endometrioid component.

The immunohistochemical profile is characterized by diffuse CKAE1/AE3 expression and aberrant nuclear β-catenin expression, typically accompanied by CDX2 positivity. CK7 and p63 positivity, as well as multifocal neuroendocrine markers expression, may be variably observed. PAX8 and ER are negative.

In Table 1, histological, immunohistochemical, and molecular features are summarized, along with the relative references.

## 4. AFP-Producing Endometrial Carcinoma (AFP-EC)

Some neoplasms, in particular hepatocellular carcinoma and yolk sac tumor, may produce alpha-fetoprotein (AFP). Interestingly, AFP is physiologically produced by the yolk sac, fetal liver, and fetal gut, and most AFP-producing tumors from different organs show morphological resemblance to these fetal structures [44,45,46,47,48].

Otani et al. reported a case series with the literature review of AFP-producing endometrial carcinoma (AFP-EC). They recognized two variants: hepatoid carcinoma and endometrial carcinoma with a fetal gut-like component; mixed hepatoid and fetal gut-like features were also described. Endometrial carcinomas with a yolk sac component were not included in the AFP+ EC category by Otani et al. [18] (*please see the next section entitled “endometrial carcinoma with yolk sac tumor differentiation”*).

Hepatoid carcinoma has been recognized as an endometrial carcinoma variant since 1996, despite not being included in the current WHO classification. Hepatoid carcinoma often shows exophytic growth and lymphovascular space invasion. It is characterized by tumor cells with prominent eosinophilic or clear cytoplasm and moderate to severe nuclear atypia; these are arranged in trabeculae with intervening sinusoid-like capillaries, imparting a striking resemblance to hepatocellular carcinoma [19,20,21].

Fetal gut-like component was characterized by tall columnar cells with clear cytoplasm arranged in glandular/papillary pattern, reminiscent of the secretory variant of endometrioid carcinoma; solid areas resembling clear cell carcinoma were also observed. Unlike hepatoid carcinoma, fetal gut-like carcinoma has not been recognized as an entity in endometrial carcinoma, but it has been reported as yolk sac tumor [18,22].

On immunohistochemistry, AFP showed variable positivity (from focal to diffuse); SALL4 (spalt-like transcription factor 4) was also expressed, at least focally, in all cases. Diffuse expression of HNF1β (hepatocyte nuclear factor 1 beta) accompanied by Napsin A negativity was observed. PAX8 and CK7 expression was partially or completely lost (while being retained in the conventional carcinoma component, when present). Estrogen and progesterone receptors were negative. Most cases were p53 abnormal; no *POLE* mutations were found [18]. Interestingly, a subset of cases showed HER2 (human epidermal growth factor receptor 2) immunohistochemical positivity, suggesting the possibility of a targeted therapy as it could be performed in serous carcinoma [18,49].

AFP+ EC appears to be aggressive as about half of the reported cases were diagnosed at advanced stage. Moreover, AFP+ EC showed aggressive behavior even at early stage [18,19,20,21]. In these tumors, serum AFP could potentially be useful in post-operative surveillance [18].

### Summary Tips for Histological Diagnosis

AFP-EC may show hepatoid (moderate amount of eosinophilic or clear cytoplasm, nuclei with coarse chromatin and moderate to severe nuclear atypia) or fetal gut-like (tall columnar cells with large nuclei and clear cytoplasm, arranged in glandular and papillary pattern) histological aspect.

The immunohistochemical profile is characterized by focal AFP, HNF1β, and SALL4 expression. CK7, PAX8, ER, and PR may be focally positive or negative. P53 usually shows aberrant expression. Napsin A and p504S are negative. CKAE1/AE3 and CK7 expression, and focal CDX2 positivity. β-catenin shows aberrant nuclear expression. PAX8 and ER are not expressed.

In Table 1, histological, immunohistochemical, and molecular features are summarized, along with the relative references.

## 5. Endometrial Carcinoma with Yolk Sac Tumor Differentiation (EC-YST)

Endometrial carcinoma with “yolk sac tumor differentiation” (EC-YST) is another rare entity which shows at least partial overlap with AFP+ EC. Yolk sac tumor in association with other somatic tumors has been more widely discussed among ovarian carcinoma [23,50]. For such tumors, the diagnostic term “somatically derived yolk sac tumors” has been proposed [23].

EC-YST seems to be very similar to its ovarian counterpart. Pathogenesis of these types of tumors was interpreted as a sort of so-called ‘‘neo-metaplasia’’, or “retro-differentiation”, originating from malignant pluripotent somatic stem cells in the context of a somatic neoplasm [23,24,50].

The few cases of EC-YST described in the literature, mostly in recent times, consist of single case reports and small case series and may arise both in childbearing age and postmenopausal women [24,25,26,27,28].

The associated somatic component is mostly represented by high-grade endometrioid carcinoma and carcinosarcoma. Morphological growth patterns of the yolk sac component are highly variable, with the most frequent patterns described consisting of microcystic-reticular patterns or glandular patterns with solid component growth; the typical endodermal (Schiller-Duval) bodies may also be present. Cytological features are highly variable and heterogeneous, including small cells with minimal cytoplasm to larger cells with cytoplasmic clearing; this last aspect must be particularly taken into account in the differential diagnosis with endometrioid carcinoma with secretory differentiation and clear cell carcinoma. Nuclear pleomorphism may be mild to moderate, with occasional severe nuclear anaplasia, and mitotic index is usually high [24,25,26,27,28].

Immunohistochemistry can help us to identify these entities: SALL4, Glypican-3 and villin are diffusely expressed in most cases; AFP and PLAP (placental alkaline phosphatase) are often expressed even if more focally. Isolated expression of one of these markers, such as SALL4, Glypican-3, and AFP is not uncommon in endometrial carcinoma without yolk sac differentiation. However, the co-expression of more than one of these markers suggests an EC-YST, although the recognition of the characteristic morphological aspects of yolk sac tumor is fundamental for definitive diagnosis [24,25,26,27,28].

Some clinically aggressive cases of EC-YST have been described in association with high-grade endometrial carcinoma. However, a prognostic and predictive value of yolk sac differentiation, independent of other clinicopathologic parameters, has not yet been highlighted by any study, because of the limited number of cases reported [24,25,26,27,28].

Whether EC-YST should be lumped together with AFP+ EC is a controversial point. Otani et al. considered the presence of yolk sac differentiation as an exclusion criterion in the diagnosis of AFP+ EC as yolk sac tumor is a germ cell tumor and thus inconsistent with a carcinoma [18]. However, as EC-YST demonstrates, the yolk sac component can be the result of a trans-differentiation of a carcinoma, as it occurs in carcinosarcoma. Moreover, the fetal gut-like differentiation is a typical feature of yolk sac tumor. In addition, AFP+ ECs assessed by Otani et al. showed co-expression of AFP and SALL4, which is considered as a feature consistent with a yolk sac component. Finally, hepatoid differentiation is not an uncommon feature in yolk sac tumor [2,22,24,25,26,27,28]. Further studies are necessary to clarify this point.

### Summary Tips for Histological Diagnosis

EC-YST is usually associated with a somatic component (high-grade endometrioid carcinoma and carcinosarcoma) and typically shows reticular, glandular, microcystic, and/or solid patterns, along with the peculiar presence of endodermal (Schiller-Duval) bodies.

The immunohistochemical profile is characterized by SALL4, Glypican-3, and villin positivity, along with focal AFP and PLAP expression.

In Table 1, histological, immunohistochemical, and molecular features are summarized, along with the relative references.

## 6. Endometrial Carcinoma with Melanocytic Differentiation (ECMD)

Melanocytic differentiation in endometrial and ovarian carcinoma is an extremely rare phenomenon, although it is likely underrecognized. In fact, melanocytes are known to be morphologically diverse, and immunohistochemical melanocytic markers are not routinely tested in endometrioid carcinoma. To the best of our knowledge, only 6 endometrial carcinomas and 3 ovarian carcinomas exhibiting a melanocytic component have been reported in the literature [37,38,39,40,41,51,52,53,54]. Two-thirds of cases were diagnosed as carcinosarcomas, while the remaining cases were serous carcinoma, endometrioid carcinoma, and neuroendocrine carcinoma, supporting that melanocytic differentiation may occur in different histotypes. Melanocytic differentiation seems to be associated with aggressive biological behavior since almost all previously published cases presented at advanced stage or developed metastases [37,38,39,40,41,51,52,53,54].

Given the morphological diversity of melanocytes, identifying melanocytic differentiation in endometrial carcinoma may be challenging. In fact, the melanocytic component in the published cases ranged from spindle cells to giant pleomorphic cells with eosinophilic cytoplasm. Two-thirds of cases showed melanin pigment, which would be of great aid in the diagnosis. All cases were positive for HMB45 (human melanoma black 45) (9/9) and Melan-A (5/5), while S100 expression was variable. Other markers found positive were Cathepsin-K, tyrosinase, and SOX10 (SRY-box transcription factor) [37,38,39,40,41,51,52,53,54].

Positivity for melanocytic markers may raise the suspicion of a PEComatous component. Immunohistochemical testing for smooth muscle markers, which are usually positive in PEComa, may be useful in the differential diagnosis between PEComatous component and melanocytic component [2,41].

In conclusion, we think pathologists should be aware of the possibility of melanocytic differentiation in endometrial carcinomas as it can be difficult to recognize but may be associated with aggressive behavior.

### Summary Tips for Histological Diagnosis

ECMDs are endometrial carcinomas exhibiting a melanocytic component, often but not always accompanied by melanin pigment.

The immunohistochemical profile is characterized by melanocytic markers positivity and frequent p53 aberrant expression. Muscle markers are negative, allowing exclusion of a possible PEComa diagnosis.

In Table 1, histological, immunohistochemical and molecular features are summarized, along with the relative references.

## 7. Endometrial Carcinoma with Histiocyte-like Tumor Cells (EC-HLTCs)

Dyscohesive tumor cells with wide eosinophilic and vacuolated cytoplasm, vaguely resembling histiocytes, can focally/multifocally be observed in endometrial carcinomas with a microcystic, elongated, and fragmented (MELF) pattern of invasion and are typically associated with acute inflammation. These “histiocyte-like tumor cells” (HLTCs) can also be found in lymphovascular tumor emboli and in lymph node metastases [55]. Similar cells may be observed in post-chemotherapy tumor specimens [56]. These HLTCs can be interpreted as tumor cells with reactive morphological features in the light of their association with inflammation or therapy.

In rare cases, endometrial carcinoma may show a prominent HLTC component in the form of diffuse sheets of HLTCs, in the absence of previous therapy. In these cases, the HLTCs appear as a distinctive tumor component rather than just reactive changes. These cells may show striking nuclear pleomorphism and sometimes signet ring-like morphology. In our previous study, we reported four cases of endometrial carcinoma with a prominent HLTC component. All cases showed a conventional carcinoma component of different histotypes (gastric-type, clear cell, serous, and gastrointestinal-type) and were at advanced stage. The HLTCs showed diffuse positivity for CK7 (which was even stronger than in the conventional carcinoma component) and PAX8. Adhesion proteins E-cadherin and β-catenin were partially altered with incomplete membranous or cytoplasmic expression. At least focal expression of CK20, HNF1β, and CK5/6 was observed in the HLTCs in all cases. However, HLTCs were negative for other gastrointestinal, squamous cell, or clear cell carcinoma markers. Apocrine, rhabdomyoblastic, and hepatoid markers were negative as well [17].

Interestingly, the immunophenotype of these HLTCs was similar to the reactive HLTCs observed in MELF cases and in post-chemotherapy carcinoma specimens. However, MELF-related HLTCs are focal/multifocal, mainly intraglandular, and associated with acute inflammation. Chemotherapy-related HLTCs may be diffuse instead; information regarding prior treatments is therefore necessary to exclude reactive changes [17].

Although the lack of cohesiveness of HLTCs may raise the question of a carcinosarcoma or a UDC/DDC, the immunophenotype of these cells shows that they retain an epithelial Müllerian differentiation. Considering that HLTCs (i) may be observed in several different histotypes, (ii) do not show evidence of trans-differentiation towards a non-Müllerian cell phenotype, and (iii) are similar to reactive cells, we suggested that they do not represent a specific entity but a “reactive-like” phenotype seemingly associated with aggressiveness [17]. Acknowledging the possibility of endometrial carcinomas with an HLTC component appears necessary to correctly classify these cases.

Current evidence is too limited to decide how to manage endometrial carcinoma with an HLTC component; however, the only currently available study suggests that HLTCs are associated with aggressive behavior as all cases were at advanced stage. This might suggest that these endometrial carcinomas should be treated as high-risk histotypes, although further studies are needed in this regard.

### Summary Tips for Histological Diagnosis

EC-HLTCs are endometrial carcinomas displaying dyscohesive cells with wide eosinophilic and vacuolated cytoplasm (i.e., histiocyte-like tumor cells).

The immunohistochemical profile is characterized by PAX8 positivity and strong and diffuse CKAE1/AE3 and CK7 expression; ER, PR, HNF1β, CK20, and CK5/6 may be variably expressed. E-cadherin and β-catenin show incomplete membrane staining and/or cytoplasmic accumulation. CDX2, p63, p40, p504s, and Napsin A are not expressed.

In Table 1 histological, immunohistochemical, and molecular features are summarized, along with the relative references.

## 8. High-Grade Corded and Hyalinized Endometrioid Carcinoma (CHEC)

Corded and hyalinized endometrioid carcinoma (CHEC) is a recognized variant of endometrioid carcinoma characterized by a corded and spindle cell component embedded in a hyaline-to-myxoid stroma; the corded component merges imperceptibly with a conventional endometrioid component. Although this biphasic pattern may raise the concern of a carcinosarcoma, the biological behavior of CHEC is analogous to conventional endometrioid carcinoma. The prototypical CHEC is a low-grade tumor, in which the corded and spindled cells show bland nuclei and low mitotic index. These low-grade features allow differentiating CHEC from carcinosarcoma [32,33,34,35,57,58,59]. However, CHECs with high-grade features have been described and may represent a serious diagnostic challenge [36].

High-grade CHEC are characterized by anastomosing cords of diffusely and markedly atypical epithelioid cells immersed in a hyaline-to-myxoid stroma, which merge imperceptibly with a high-grade endometrioid component. Mitotic activity is evident in the corded cells, although it may be lower than in the endometrioid component. Similar to prototypical low-grade CHEC, squamous differentiation is invariably present, and keratinizing features are often observed within single cells or small cell clusters in the corded component. The stroma may show osteoid or chondroid changes. Interestingly, the corded component is usually located at the surface of the tumor and non-myoinvasive.

Immunohistochemically, the corded cells show decreased expression of epithelial markers and hormone receptors and frequent nuclear accumulation of β-catenin. Unlike low-grade CHEC, high-grade CHEC often shows MMR-deficiency or p53-abnormal expression [35,36].

Although high-grade CHECs may resemble carcinosarcoma or DDC, they show peculiar distinctive features. The most striking of these features obviously is the morphology of the corded component. Such component differs from the undifferentiated component of DDC, which is characterized by solid sheets of dyscohesive monomorphic cells showing a sharp demarcation with a conventional carcinoma component (which is typically low-grade). Moreover, UDC/DDC often show loss of expression of SWI/SNF proteins, i.e., ARID1B, SMARCA4/BRG1, and/or SMARCB1/INI1. On the other hand, carcinosarcoma often shows a serous carcinoma component and an overt sarcomatous component and does not show nuclear β-catenin accumulation. In both UDC/DDC and carcinoma, the non-epithelial component is often diffusely myoinvasive [2,12,13,14,15,16,35,36,42].

With regard to prognosis, it is reasonable to hypothesize that, if low-grade CHEC behaves similarly to low-grade endometrioid carcinoma, then high-grade CHEC behaves similarly to FIGO G3 endometrioid carcinoma. Therefore, it might be expected that its prognosis is affected by the TCGA molecular group and other relevant histopathological factors such as depth of myometrial infiltration and lymphovascular space invasion. Further studies are necessary to clarify this point.

### Summary Tips for Histological Diagnosis

CHECs are characterized by the presence of an endometrioid component merging with a corded component, showing bland appearance and low mitotic activity. Squamous/morular metaplasia is usually prominent.

The immunohistochemical profile is characterized by decreased expression of CKAE1/AE3 and E-cadherin in the corded component, along with β-catenin aberrant nuclear expression. ER and PR are usually negative or only focally expressed in the corded component.

In Table 1, histological, immunohistochemical and molecular features are summarized, along with the relative references.

## 9. Discussion

### Ultra-Rare Histological Variants: A Priori Aggressive Histotypes in the 2023 FIGO Staging System?

Many important advances in the understanding of the pathologic features, precursor lesions, molecular background, and natural history of endometrial cancer have occurred since the 2009 FIGO staging system. More data regarding clinical outcome, genetic landscape, and biological behavior are now available regarding the several common histological types [42].

Considering that the histological tumor type is an important prognostic predictor in endometrial carcinoma and it is used to guide treatment together with grade and stage, the histopathological findings have a central role in the recent 2023 revision of the FIGO staging of endometrial carcinoma [60].

Defining the place of rare or emerging variants of endometrial carcinoma in the current risk stratification system appears of paramount importance. In fact, although any of these entities is rare, they may constitute a clinically relevant proportion of endometrial carcinomas if taken together. In our experience, it is not uncommon that pathologists (even those with experience in gynecological pathology) unsuccessfully try to ascribe these entities to any of the histotypes described by the WHO. When different pathologists review these cases, they often make different diagnoses, none of which actually fits the specific case. This creates confusion for both patients and clinicians regarding the correct management.

The most immediate solution would be to consider all these uncommon variants as *a priori* aggressive histotypes. However, while such an approach appears reasonable for some carcinomas such as EGCC (given its anaplastic morphology), it is not obvious for others such as high-grade CHEC (which might behave as an endometrioid carcinoma despite resembling carcinosarcoma). It remains to be defined whether some of the described entities are morphological variants of other carcinomas or distinct entities requiring different treatment. It is also unclear how other prognostic parameters, such as depth of myometrial invasion, lymphovascular space invasion, and molecular class, impact their biological behavior. For instance, it has been suggested that mismatch repair deficiency is associated with better prognosis in carcinosarcoma but not in UDC/DDC [42,61,62]. In this regard, Weisman et al. suggested that mismatch repair deficiency does not improve the prognosis of PiMHEC [63].

## 10. Conclusions

To conclude, nearly all the types of endometrial carcinomas discussed in this review might be better considered, at the state of the art, as rare variants of the established categories, rather than independent rare types. However, some peculiar features may also suggest that some of these variants may represent distinct entities, potentially showing a different prognosis thus requiring a personalized therapeutic approach. By better identification and characterization of these EC variants, clinicians can leverage existing treatment protocols and adapt them to the specific characteristics of these variants or approach them as entirely novel entities. This could lead to more accurate diagnoses, informed therapeutic decisions, and effective management strategies tailored to the unique behavior of each variant/entity. Increased awareness and recognition of these variants might also facilitate earlier interventions and better prognostic assessments, ultimately contributing to improved survival rates and quality of life for patients diagnosed with endometrial carcinoma. We hope that our review may promote a more thorough and systematic study of these rare entities, through multicentric collaborations, in order to better define their diagnostic criteria, clinicopathological features, biological behavior, and treatment.

## Figures and Tables

**Table 1 ijms-25-10320-t001:** Summary of histological, immunohistochemical, and molecular features of rare EC variants, along with relative references and number of cases reported.

Rare Variants	Histological Features	Immunohistochemistry	Molecular Alterations and TCGA Signature	References	Number of Cases
Endometrial Giant Cell Carcinoma (EGCC)	- Giant cells with marked cytological atypia and extreme pleomorphism - Sheets of anaplastic, mono- or multi-nucleated cells	- CKAE1/AE3: focal/multifocal - EMA: focal/multifocal - Vimentin: positive (50%) - E-cadherin: negative - β-HCG: negative - Muscle markers: negative	- NSMP preferentially (*currently limited data*)	[3]	5
				[4]	6
				[5]	3
				[6]	1
				[7]	1
				[8]	1
				[9]	1
				[10]	1
					Total: 19
Endometrial carcinoma with histiocyte-like tumor cells (EC—HLTCs)	- Dyschoesive cells with wide eosinophilic and vacuolated cytoplasm (histiocyte-like tumor cells, HLTCs)	- CKAE1/AE3: diffuse and strong - CK7: diffuse and strong - CK20: multifocal/diffuse - CK5/6: multifocal - E-cadherin: incomplete membrane staining and/or cytoplasmic accumulation - β-catenin: incomplete membrane staining and/or cytoplasmic accumulation - PAX8: positive - ER: focal/negative - PR: focal/negative - CDX2: negative - p63 and p40: negative - HNF1β: positive - Napsin A and p504S: negative - p53: aberrant or wild-type expression	- MMRd or p53abn (*currently limited data*)	[17]	4
					Total: 4
AFP-producing endometrial carcinoma (AFP-EC)	- Hepatoid: moderate amount of eosinophilic or clear cytoplasm, nuclei with coarse chromatin and moderate to severe nuclear atypia - Fetal gut-like: tall columnar cells with large nuclei and clear cytoplasms arranged in glandular and papillary pattern	- AFP: focal - SALL4: focal - CK7: focal/negative - PAX8: focal/negative - ER: focal/negative - PR: focal/negative - HNF1β: positive - Napsin A and p504S: negative - p53: frequent aberrant expression (80%)	- p53abn (*currently limited data*)	[18]	5
				[19]	1
				[20]	1
					Total: 7
Endometrial carcinoma with yolk sac tumor differentiation (EC-YST)	- Association to a somatic component (high grade endometrioid carcinoma and carcinosarcoma) - Reticular, glandular microcystic, solid patterns - Presence of the typical endodermal (Schiller-Duval) bodies - Cytological heterogeneity	- SALL4, Glypican-3 and villin: positive - AFP and PLAP: focally positive	(*currently limited data*)	[21]	1
				[22]	9
				[23]	1
				[24]	1
				[25]	1
				[26]	2
				[27]	1
				[28]	1
					Total: 17
Pilomatrix-like high-grade endometrioid carcinoma (PiMHEC)	- Solid growth pattern with a neoplastic proliferation of high-grade basaloid cells associated with diffuse ghost cell keratinization and extensive geographic necrosis - In most cases of PiMHEC exhibit a low-grade endometrioid component	- CKAE1/AE3: diffuse - CK7: variable - CDX2: focal - PAX8: negative - ER: negative - β-catenin: aberrant nuclear expression	- CTNNB1 mutations - NSMP (*currently limited data*)	[29]	5
				[30]	1
				[31]	1
					Total: 7
Corded and hyalinized endometrioid carcinoma (CHEC)	- Corded component: bland appearance and low mitotic activity - Merging of the corded and endometrioid components - Prominent squamous/morular metaplasia	- CKAE1/AE3: decreased expression in the corded component - E-cadherin: decreased expression in the corded component - β-catenin: aberrant nuclear expression - ER and PR: focal/negative	- NSMP (*currently limited data*)	[32]	31
				[33]	5
				[34]	9
				[35]	62
				[36]	6
					Total: 113
*Endometrial carcinoma with melanocytic differentiation* *(ECMD)*	- Endometrial tumors exhibiting a melanocytic component - Presence of melanin pigment	- Melanocytic markers (S100, Melan A, HMB45): positive - Muscle markers: negative - p53: frequent aberrant expression	- p53abn (*currently limited data*)	[37]	1
				[38]	1
				[39]	1
				[40]	1
				[41]	1
					Total: 5

**Table 2 ijms-25-10320-t002:** Summary of useful tips for differential diagnosis of PiMHEC mimickers.

Histological Type	Useful Tips for Differential Diagnosis vs. PiMHEC
High-grade endometrioid EC with solid pattern and squamous features	- lacks ghost cell keratinization - no aberrant nuclear expression of β-catenin - at least focal expression of Müllerian markers
Low-grade endometrioid EC with prominent ghost cell keratinization	- keratinization is typically associated with morular metaplasia - admixture of squamous and glandular elements (highlighted by mixed membranous and nuclear β-catenin expression, and by patchy positivity for Müllerian markers)
Serous EC with solid growth pattern	- no keratinization - aberrant p53 expression
UDC/DCC with diffuse nuclear β-catenin accumulation and areas of abrupt keratinization	- the undifferentiated cells are dyscohesive and organized in patternless sheets - no ghost cell keratinization
NEC	- no ghost cell keratinization - diffuse positivity for neuroendocrine markers
Squamous cell carcinoma	- very rare as pure histological type in endometrium - no ghost cell keratinization - no aberrant nuclear expression of β-catenin

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
