# Peer review of "Beyond the WHO 2020 Classification of Female Genital Tumors: Types of Endometrial Cancer: A Pathological and Molecular Focus on Challenging Rare Variants"

_ijms, 2024, doi:10.3390/ijms251910320_

Round 1

Reviewer 1 Report

Comments and Suggestions for Authors

The present review article is aimed to present clinico-pathological and molecular landscapes of rare endometrial cancer variants along with the discussion of their differential diagnosis and management course. Knowledge on rare variants of endometrial cancer (EC) is limited. These variants are not included in the 2020 World  Health Organization (WHO),  presenting a real diagnostic challenge for Pathologists.  It is stated that it is vital to perform multicenter and prospective studies in order to elucidate their molecular landscape and for new therapeutic windows.  It is suggested that a better histological typing and characterization of these EC variants may lead to more reproducible and accurate diagnoses for better personalized treatments.

Given the aim and focus, the review is portraying eight rare variants of EC. The review  is  poorly written, mainly presented as a collection of phenotype-histological descriptions. No decisive conclusions per variant are provided. It rather uses a slang language probably familiar to ovarian /endometrial cancer people and contains many abbreviations  which are not familiar to the general readers.

The main hurdle is that no multi center data are presented nor the number of population for each variant or total cases studied. 

Specific points:

-Figure 1. The figure is unacceptable. It contains a collection of different variant subtypes all of which but one are H&E staining. Showing the different H&E variants should be compared with normal EC epithelial H&E.

A different figure should present IHC (immune histo-chemistry) staining of at least one typical marker of a given variant. This will clarify and help the readers to follow the descriptive paragraphs per variant described. 

- A summary of pivotal conclusion per each variant should be presented.

- Tables should include number of cases studied.

-Line : 37 : what is ppen?

- Lines 139-141: is not a sentence. Probably should be divided to two sentences.

- Abbreviations should be explained. For example: HNF1b, POLE, and more…

Minor: e-cadherin should be : E-cadherin

Comments on the Quality of English Language

English editing by a  native speaking English person is recommended.

Author Response

Rev 1

Comments and Suggestions for Authors

The present review article is aimed to present clinico-pathological and molecular landscapes of rare endometrial cancer variants along with the discussion of their differential diagnosis and management course. Knowledge on rare variants of endometrial cancer (EC) is limited. These variants are not included in the 2020 World Health Organization (WHO), presenting a real diagnostic challenge for Pathologists. It is stated that it is vital to perform multicenter and prospective studies in order to elucidate their molecular landscape and for new therapeutic windows. It is suggested that a better histological typing and characterization of these EC variants may lead to more reproducible and accurate diagnoses for better personalized treatments.

Given the aim and focus, the review is portraying eight rare variants of EC. The review is poorly written, mainly presented as a collection of phenotype-histological descriptions. No decisive conclusions per variant are provided. It rather uses a slang language probably familiar to ovarian /endometrial cancer people and contains many abbreviations which are not familiar to the general readers.

The main hurdle is that no multi center data are presented nor the number of population for each variant or total cases studied. 

Specific points:

-Figure 1. The figure is unacceptable. It contains a collection of different variant subtypes all of which but one are H&E staining. Showing the different H&E variants should be compared with normal EC epithelial H&E.

A different figure should present IHC (immune histo-chemistry) staining of at least one typical marker of a given variant. This will clarify and help the readers to follow the descriptive paragraphs per variant described. 

We have modified the figures according to your precious suggestion

- A summary of pivotal conclusion per each variant should be presented.

We greatly thank the Reviewer for this suggestion. As suggested, we inserted a summary paragraph per each variant, describing the essential tips for an appropriate histological diagnosis.

- Tables should include number of cases studied.

We thank the Reviewer for her/his suggestion, which allowed us to improve our manuscript. We modified the table 1 accordingly, including references for all the cases reported in literature and also the number of cases studied.

- Line: 37: what is ppen?

We are sorry, it was a misspelling for “open”.

- Lines 139-141: is not a sentence. Probably should be divided to two sentences.

We are sorry, it was a mistake. It has now been separated into two sentences.

- Abbreviations should be explained. For example: HNF1b, POLE, and more…

Amended, thank you

Minor: e-cadherin should be: E-cadherin

Amended, thank you

Reviewer 2 Report

Comments and Suggestions for Authors

Dear authors,

Thanks for your outstanding work. It is really unique and very well-written

Only a few remarks to be noted:

1- The abstract is likely an introduction rather than an abstract. It needs to be rewritten as an abstract

2- Please divide the figure so that every figure follows its type

3- Mimickers of PiMHEC are better to be collected in one Table

4- Nearly all the types can be considered rare variants of the known types rather than independent rare types. Please discuss how your proposal can affect the patient management and outcomes

Author Response

Rev 2

Dear authors,

Thanks for your outstanding work. It is really unique and very well-written

We greatly thank the Reviewer for her/his appreciation.

Only a few remarks to be noted:

1- The abstract is likely an introduction rather than an abstract. It needs to be rewritten as an abstract

We thank the Reviewer for her/his advice. The abstrasct has been completely modified, according to her/his suggestion.

2- Please divide the figure so that every figure follows its type

We have modified the figures section

3- Mimickers of PiMHEC are better to be collected in one Table

We thank the Reviewer for her/his suggestion. We now included Table 2, which summarizes features of PiMHEC mimickers.

4- Nearly all the types can be considered rare variants of the known types rather than independent rare types. Please discuss how your proposal can affect the patient management and outcomes.

We thank the Reviewer for her/his suggestion. We included a “conclusion” paragraph, discussing this point.

Reviewer 3 Report

Comments and Suggestions for Authors

The manuscript presents detailed and valuable characterization of rare variants in EC. It is well written, data presented is clear. 

Minor:

- I would recommend showing more clinical data on tumor behavior, and follow-up of patients (overall survival, tendency to recur, or suitable types of management )

- SEC type is mentioned in the table but it is not well described in the text

Comments on the Quality of English Language

I would consider some small editting changes to the highlights section. 

The manuscript is well written, but please read the text carefully for minor mistakes. 

Author Response

Rev 3

Comments and Suggestions for Authors

The manuscript presents detailed and valuable characterization of rare variants in EC. It is well written, data presented is clear. 

Minor:

- I would recommend showing more clinical data on tumor behavior, and follow-up of patients (overall survival, tendency to recur, or suitable types of management )

We thank the Reviewer for her/his suggestion. We completely agree, but unfortunately there is a lack of clinical data as regards behavior and follow up of these rare entities/variants. We hope that our manuscript will improve their awareness and recognition, in order of collecting more clinical data in the next future. We also included a paragraph discussing this point at the end of the manuscript.

- SEC type is mentioned in the table but it is not well described in the text

We are sorry, it was a mistake. SEC type is not included in the text and has now been eliminated also from the table.

Comments on the Quality of English Language

I would consider some small editting changes to the highlights section. 

We thank the Reviewer for her/his suggestion. The highlights section has been modified accordingly.

The manuscript is well written, but please read the text carefully for minor mistakes. 

We thank the Reviewer for her/his suggestion. The text has now been carefully revised.

Round 2

Reviewer 1 Report

Comments and Suggestions for Authors

Unfortunately, the revised manuscript by Santoro A et al., is still poorly written. It is overloaded with ample histological details, mainly descriptive which makes it difficult to comprehend especially for the innocent reader.

Many errors ad misspelled items appear throughout the manuscript.

For example:

-Fig. 1B what is:  multiocal ?

- In the legends for figures it should state Fig. 2 etc for all other figures.

-Line 217: what is exophytic?

--Line 244: "The immunohistochemical profile is characterized by focal AFP, HNF1β and SALL4 expression. CK7, PAX8, ER, PR may be focally positive or negative." What is the message here, not clear!

-Line 246: "CKAE1/AE3 and CK7 expression, and focal CDX2 positivity."  This is NOT a sentence.

References: 9, 10, 40-46, 50, 51; 10 references out of the 63  are written in a different style!

This is rather  sloppy which points  on the entire gravity of the present review manuscript.

- Significantly, extensive editing of the English language by a native English speaker is required. The text is difficult to understand and in its present form is incomprehensible.

Comments on the Quality of English Language

The review manuscript in its present form is inappropriate for publication. Extensive English editing is needed by a native English speaking individual !!!

Author Response

Response to Reviewer 1

Comment 1: Unfortunately, the revised manuscript by Santoro A et al., is still poorly written. It is overloaded with ample histological details, mainly descriptive which makes it difficult to comprehend especially for the innocent reader. 

Response 1: We thank the reviewer for its comments. We tried to simplify the manuscript as suggested. However, its important top point out that  our paper aims to describe all histological details related to endometrial cancer histotypes in order to help pathologists to recognize rare and emerging variants of endometrial carcinoma.

Comment 2: Many errors ad misspelled items appear throughout the manuscript.  -Fig. 1B what is:  multiocal ?

Response 2: We have corrected; the correct term is multifocal

Comment 3: - In the legends for figures it should state Fig. 2 etc for all other figures.

Response 3: We have corrected

Comment 4: -Line 217: what is exophytic?

Response 4: Exophytic is a specific term used in pathology, referred to tumors tending to grow outward beyond the surface epithelium from which it originates. We have included this explanation within text.

Comment 5: --Line 244: "The immunohistochemical profile is characterized by focal AFP, HNF1β and SALL4 expression. CK7, PAX8, ER, PR may be focally positive or negative." What is the message here, not clear!

Response 5: As also done in other paragraphs, our intention was to provide a summarized list of histological and immunohistochemical features. As stated in line 244, for this specific histological subtype, the immunohistochemical markers CK7, PAX8, ER and PR may be focally expressed (positive) or may be not expressed (negative).

Comment 6: -Line 246: "CKAE1/AE3 and CK7 expression, and focal CDX2 positivity."  This is NOT a sentence.

Response 6: We apologize with the reviewer, the sentence has been deleted and the paragraph is correct now.

Comment 7: References: 9, 10, 40-46, 50, 51; 10 references out of the 63 are written in a different style!

Response 7: as suggested, all references have been modified according to the journal guidelines.

Comment 8:  Significantly, extensive editing of the English language by a native English speaker is required. The text is difficult to understand and in its present form is incomprehensible. 

Response 8: We thank the Reviewer for the suggestions and for all the precious advices that allowed us to improve our manuscript. The text has been now fully revised, with the help of a native English speaker. We would also like to clarify that our review is mainly directed to pathologists. We apologize if it is overloaded with ample histological details, but we believed it was fundamental in order to explain the histological appearance of these extremely rare entities and to give the correct instruments in order to allow pathologists to recognize and diagnose these histological subtypes.

Round 3

Reviewer 1 Report

Comments and Suggestions for Authors

The review article by Santoro A et al.,   is now suitable for publication.

The short summary at the end of each paragraph is helpful. Correction of misspelled items and miss leading term expression make it better to comprehend.  Yet, the article is certainly suitable for a specifically specialized type of readers.